# Invasiveness of *Escherichia coli* Is Associated with an IncFII Plasmid

**DOI:** 10.3390/pathogens10121645

**Published:** 2021-12-20

**Authors:** Lars Johannes Krall, Sabrina Klein, Sébastien Boutin, Chia Ching Wu, Aline Sähr, Megan L. Stanifer, Steeve Boulant, Klaus Heeg, Dennis Nurjadi, Dagmar Hildebrand

**Affiliations:** 1Department of Infectious Diseases, Medical Microbiology and Hygiene, Heidelberg University Hospital, 69120 Heidelberg, Germany; larsjohanneskrall@gmail.com (L.J.K.); sabrina.klein@med.uni-heidelberg.de (S.K.); sebastien.boutin@med.uni-heidelberg.de (S.B.); chiachingwu1993@gmail.com (C.C.W.); Aline.Saehr@med.uni-heidelberg.de (A.S.); Klaus.Heeg@med.uni-heidelberg.de (K.H.); dennis.nurjadi@med.uni-heidelberg.de (D.N.); 2DZIF German Center for Infection Research, 38124 Braunschweig, Germany; 3Department of Infectious Diseases, Molecular Virology, Heidelberg University Hospital, 69120 Heidelberg, Germany; m.stanifer@dkfz-heidelberg.de; 4Department of Infectious Diseases, Virology, Heidelberg University Hospital, 69120 Heidelberg, Germany; s.boulant@Dkfz-Heidelberg.de; 5Research Group “Cellular Polarity and Viral Infection”, DKFZ, 69120 Heidelberg, Germany

**Keywords:** *Escherichia coli*, *hha*, bloodstream infection, invasion, gut organoid model, IncFII plasmid

## Abstract

*Escherichia coli* is one of the most prevalent pathogens, causing a variety of infections including bloodstream infections. At the same time, it can be found as a commensal, being part of the intestinal microflora. While it is widely accepted that pathogenic strains can evolve from colonizing *E. coli* strains, the evolutionary route facilitating the commensal-to-pathogen transition is complex and remains not fully understood. Identification of the underlying mechanisms and genetic changes remains challenging. To investigate the factors involved in the transition from intestinal commensal to invasive *E. coli* causing bloodstream infections, we compared *E. coli* isolated from blood culture to isolates from the rectal flora of the same individuals by whole genome sequencing to identify clonally related strains and potentially relevant virulence factors. in vitro invasion assays using a Caco- 2 cell intestinal epithelial barrier model and a gut organoid model were performed to compare clonally related *E. coli.* The experiments revealed a correlation between the presence of an IncFII plasmid carrying *hha* and the degree of invasiveness. In summary, we provide evidence for the role of an IncFII plasmid in the transition of colonization to invasion in clinical *E. coli* isolates.

## 1. Introduction

*Escherichia coli* is one of the major pathogens causing hospital-acquired and community-acquired infections such as pneumonia, urinary tract (UTI), and bloodstream infections (BSI) [1]. At the same time, *E. coli* is part of the microbial flora of the healthy human gut. It is widely accepted, that infections by *E. coli* are mainly of endogenous origin and may be a consequence of translocation from the gastrointestinal tract [1,2].

Several virulence factors influencing adhesion, invasion, and biofilm formation, as well as their regulatory networks, have been described for pathogenic *E. coli*. These factors and their regulators, as well as antimicrobial resistance (AMR) genes, can be found also on mobile genetic elements such as plasmids, which can be transferred via horizontal gene transfer (HGT) between different bacterial strains and species [3].

One important regulatory player in the regulation of virulence factors in *E. coli* is the hemolysin expression modulating protein (Hha). Hha is a small (8-kDa) protein, found among enteric bacteria, that enhances repression of a subset of H-NS (histone-like nucleoid structuring) regulated genes [4]. This negative regulation of gene expression mostly applies to newly acquired sequences (e.g., via HGT) and might prevent potentially harmful effects of their uncontrolled expression [5]. Hha has been shown to modulate biofilm formation, and flagellar and curli gene expression through transcriptional regulation in *E. coli* [6,7]. It was also found on plasmids carrying AMR genes in enterohemorrhagic (EHEC) and enteropathogenic (EPEC) *E. coli* [8]. Therefore, *hha* can be transferred via HGT along with the acquisition of AMR genes.

The aim of this study was to investigate factors involved in the transition of colonizing *E. coli* isolates to invasive strains. In order to do so, we identified *E. coli* blood culture isolates and clonally related isolates from rectal swabs of the same individuals through whole genome sequencing (WGS). The invasive properties of the isolates were investigated using an in vitro invasion assay. We identified an IncFII plasmid carrying *hha* associated with the invasiveness of one *E. coli* isolate. Removal of the plasmid abolished the invasive properties in vitro.

In conclusion, our findings suggest that the transfer of a resistance plasmid may contribute to the transition of commensal to invasive *E. coli* strains in vivo.

## 2. Results

### 2.1. Clinical Isolates and Molecular Characterization

We identified five patients in the study period with bloodstream infections (BSIs) caused by *E. coli* (Eco_b) and a corresponding rectal isolate (Eco_r). All 10 isolates were sequenced to determine their clonal relationship. Phylogeny based on core genomes (3386 genes) indicated that in 4/5 (80%) of *E. coli* BSI cases, the rectal isolates were genetically closely related to the *E. coli* isolate from the blood culture (Figure 1a). BSI and rectal isolates from P4 were genetically not related.

Interestingly, isolates of three patients (P1, P4 and P5) carried an extrachromosomal variant of hha, but those of P2 and P3 did not. Both isolates of P5 harbored *hha*, whereas of P1, only the blood culture isolate was *hha*-positive (Figure 1a). Characterization of the *hha* genes revealed the presence of two different extrachromosomal variants (Figure 1b). P1 harbored the extrachromosomal *hha* variant 1, which was located on an IncFII plasmid, while P5 harbored *hha* variant 2 located on an IncX5 plasmid. 

### 2.2. In Vitro Invasiveness of Clinical E. coli Isolates

As Hha is a regulatory protein that is known to influence the virulence factors in *E. coli*, we aimed to study the invasive properties of clinical isolates carrying an extrachromosomal *hha* variant in the context of the gut epithelial barrier invasion. We performed an in vitro infection assay using a human gut epithelial cell line (CaCo-2) in a transwell system. Isolates of P1 and P5 were chosen for further analysis, as they carried an extrachromosomal *hha* variant and the blood culture isolate was genetically related to the rectal isolate. Isolates of P2-P4 were excluded, as those of P4 were genetically unrelated and those of P2 and P3 did not carry an extrachromosomal *hha* variant (Figure 1a). Both isolates of P5 (Eco_b5 and Eco_r5), carrying an extrachromosomal *hha* variant, did not show significant differences in the epithelial invasion (Figure 2a) or transmigration (Figure 2B) between the blood culture and rectal isolates. In contrast, the *hha*^+^ blood culture isolate of P1 (Eco_b1) was significantly invasive in the CaCo-2 model, as demonstrated by higher colony density (CFU/mL), compared to the *hha*^−^ rectal isolate (Eco_r1; Figure 2 a). Epithelial penetration of the *hha*^+^ Eco_b1 was also increased, although not statistically significant (Figure 2b). 

### 2.3. Loss of Function by Plasmid Curing

Since invasion experiments suggested that the *hha*^+^ Eco_b1 was more invasive than the genetically related *hha*^−^ Eco_r1, we next investigated whether loss of the *hha*-harboring plasmid in Eco_b1 would reduce the invasiveness of this isolate. To do so, we cured the blood culture isolate Eco_b1 of P1 (Figure 3a and Appendix A) from the plasmid and repeated the in vitro invasion experiments. Plasmid curing of the blood culture isolate reduced the invasiveness in the in vitro model significantly (*p* = 0.01; Figure 3b,c).

### 2.4. Confirmation of Findings in a Gut Organoid Model

To confirm the findings of the in vitro transwell model using the CaCo-2 cell line in a more physiological system, invasion experiments were repeated in a gut organoid model. As in the gut organoid model, *hha*+ Eco_b1 was significantly more invasive (*p* = 0.03) than *hha*- Eco_r1. The cured *hha*^−^ blood culture isolate was less invasive (*p* = 0.03, Figure 3d) than *hha*^+^ Eco_b1 and comparable to *hha*^−^ Eco_r1, suggesting a dependency of invasiveness on the plasmid content. 

### 2.5. Plasmid Transfer into J53 E. coli

Since the *hha* gene is located on a transferable resistance plasmid, we next investigated if the acquisition of this plasmid in a different *E. coli* genetic background would increase the invasiveness of a J53 recipient isolate poor in chromosomal encoded virulence factors. 

First, we compared the virulome of the *E. coli* strains from P1 (Eco_b1, Eco_r1) and J53 (Appendix A). As expected, the virulence profile of Eco_b1 and Eco_r1 was different than that of J53. In Eco_b1 and Eco_r1, a range of genes encoding for virulence factors affecting iron transport and metabolism (chuA, chuS, chuT, chuU, chuW, chuX, chuY, fyuA, irpa, irp2, iucA, iucB, iucC, iucD), secretion and transport (gspD, gspE, gspF, gspG, gspH, gspI, gspJ, gspK, kpsD, kpsM, sat), adhesion (fdeC, papI, papX) and transcription regulation (ybtA, ybtE, ybtP, ybtQ, ybtS, ybtT, ybtU, ybtX) were present, but were absent in J53. In J53, genes encoding for factors involved in a type III secretion system were present (espL1, espL4, espR1, espX4, espX5, espY4) and for expression of fimbriae (fimB), but were missing in Eco_b1 and Eco_r1. Then, we transferred the IncFII plasmid into the *E. coli* laboratory strain J53 by liquid mating. Although plasmid transfer into J53 was successful, as confirmed by detection of hha by WGS, invasiveness was not enhanced in the J53 bearing the hha^+^ IncFII plasmid (Figure 4). This strengthened the hypothesis that Hha regulates chromosomal encoded virulence factors.

## 3. Discussion

In this study, we aimed to identify clonally related clinical *E. coli* isolates from blood cultures and the rectal flora to investigate potential mechanisms involved in, or affecting, the transition of commensal to invasive *E. coli*.

In the isolates included in the study, we detected extrachromosomal *hha* in four of ten analyzed *E. coli* isolates, with two different genetic variants. The *Hha* variant 1 was detected in Eco_b1 and Eco_b4. When comparing the invasive properties of the Eco_b1 carrying *hha* variant 1 on an IncFII plasmid with Eco_r1, we observed a marked difference in the invasion into epithelial cells. Having selected in vivo invasive isolates from BSI, we were able to employ an intestinal cell culture, as well as a gut organoid model, and study the invasiveness ex vivo. As Eco_b1 and Eco_r1 are clonally related and possess identical virulence genes, regulatory effects of Hha may explain the differences in invasive properties. In Eco_b5 and Eco_r5, the *hha* variant 2 was detected in both isolates and the isolates were clonally related. The invasion assay did not show significant differences in invasiveness. There are two explanations for this observation. First, there may be differences related to the two different *hha* gene variants present in isolates of P1 and P5. Second, the genetic background in terms of virulence genes may be different and the regulatory effect of Hha did not result in enhanced invasion in isolates of P5. However, as we investigated clinical blood culture isolates, the ability to invade into the bloodstream was a prerequisite of all isolates. Thus, one may have expected a higher capability to invade intestinal epithelial cells. On the other hand, individual host factors may contribute to the success of bloodstream invasion [2] that are independent of, or complement the characteristics of the pathogen.

Our findings point to a dependency of invasion in intestinal epithelial cells on the presence of a plasmid carrying *hha*. The *hha* gene-encoded protein belongs to the Hha family of nucleoid associated proteins [9]. Hha proteins are involved in the modulation of virulence gene expression in response to environmental cues such as temperature, salt, and pH, and are known to interact with and fine tune H-NS proteins to silence gene expression [10]. *Hha* genes can be either chromosomally or plasmid-encoded, and the structure slightly differs between the chromosomal and plasmid variant. IncF plasmids can carry *hha* encoding Hha without hns [11], supporting the hypothesis that plasmid Hha might only regulate genes carried on the plasmid itself. At least in *Salmonella typhimurium*, an H-NS-independent Hha-conducted regulation of genes could be observed [12].

Several studies on Hha regulating bacterial pathogenic factors have been published. Most of them concentrate on expression profiling of known pathogenic factors involved in alpha-hemolysin expression, adherence and biofilm formation [6,7,13]. Many publications display a negative regulation, not giving a hint for an Hha-mediated increased virulence.

However, Ren et al. reported that Hha is upregulated in *E.coli* biofilm compared to suspension cells [14], and *hha* deletion was shown to attenuate aggregation and biofilm formation [13]. A study from Pietro et al. found chromosomal *hha* and two paralogous *hha2* and *hha3* predominately in some groups of intestinal pathogenic *E. coli* strains (enteroaggregative and Shiga toxin-producing isolates), as well as in the widely distributed extraintestinal ST131. The authors state a correlation between all three *hha* alleles with virulence phenotype of respective *E. coli* strains [15]. Furthermore, in Salmonella, *hha* mutants exhibit an initial hyperinvasive phenotype in cell culture but are attenuated for virulence in competitive murine infection models [16,17].

In line with this finding, we showed that *E.coli* harboring a *hha*-encoding IncFII_1 plasmid display enhanced invasive properties into intestinal epithelial cells compared to clonal *E.coli* isolates without the plasmid. 

Our study has limitations. As we analyzed 10 clinical bacterial isolates, we cannot conclude on a general pathogenic mechanism or on the frequency of this event. This is underlined by the fact that *hha* was not present in all of the *E. coli* isolates. In addition, we randomly selected *E. coli* blood culture isolates identified during routine patient care when a corresponding rectal isolate was present. We did not record information on clinical presentation or underlying diseases and cannot comment on the role of host factors such as immunosuppression, as this may also affect the invasiveness of the investigated isolates in vivo, resulting in bloodstream infection [2]. However, we provide evidence that the presence of an IncFII plasmid carrying *hha* is responsible for enhanced invasiveness in an *E. coli* blood culture isolate and that removal of the plasmid results in loss of invasive properties. However, although hha is the most likely candidate, we cannot exclude other genes present on the plasmid to be involved in virulence factor regulation.

In conclusion, we deployed a novel investigation approach comparing and analyzing clonal *E. coli* isolates from rectal swabs and blood cultures of the same individuals. We conclude that the presence of an IncFII_1 plasmid, gained by horizontal gene transfer, to contribute to increased invasiveness and virulence, and that this is one exemplary mechanism in the transition of commensal to invasive *E. coli*. This finding contributes to the understanding of the complex mechanisms and factors that affect the transition from commensal to invasive strains and is especially interesting regarding the potential dissemination of virulence through HGT. 

## 4. Materials and Methods

### 4.1. Study Premise and Clinical E. coli Isolates

Patients treated at Heidelberg University Hospital in the period of May 2018 until September 2019 presenting with bloodstream infection caused by *E. coli* and a corresponding rectal swab with cultural detection of *E. coli* were included in the study. Bacterial isolates were cryopreserved for further analyses. The Ethical Review Board of the University of Heidelberg approved the study protocol and waived individual informed consent (S-474/2018). Baseline characteristics of the patients whose isolates were included in the study are displayed in Appendix A.

### 4.2. Whole Genome Sequencing and Data Analyses

DNA extraction, library preparation and sequencing on a MiSeq Illumina platform (short-read sequencing 2 × 300 bp) were performed as previously described [18]. In brief, raw sequences were controlled for quality using sickle (v1.33, parameter: −q 30 −l 45), assembled with SPAdes 3.13.0 (with the option –careful and –only-assembler). A draft genome was curated by removing contigs with a length <1000 bp and/or coverage <10×. The quality of the final draft was quality controlled using Quast (v5.0.2). Annotation was performed with Prokka 1.14.1 and the core genome was calculated using Roary 3.12 (gene present in all the isolates). The complete draft genomes were processed through available databases using Abricate to identify virulence factors (VFDB database), antimicrobial resistance (NCBI, CARD, ARG-ANNOT, Resfinder, MEGARES databases) and plasmid type (PlasmidFinder database).

### 4.3. Data Availability, Accession and Sequence Statistics

The draft genome sequences are deposited in the NCBI GenBank database under the project number PRJNA745586. Accession numbers and sequence statistics are displayed in Table 1.

### 4.4. Invasion and Permeation Assay

To analyze invasiveness of *E. coli* clinical isolates, the colonic immortalized enterocyte cell line (CaCo-2) was used. Cells were cultivated in DMEM (anprotec, Germany) with 20% FCS (anprotec), 1% penicillin/streptomycin (anprotec) and 1% non-essential amino acids (Sigma Aldrich, Merck, Germany) in a collagen-coated transwell system (Corning, NY, USA) under 1% O_2_ and 5% CO_2_ for 21 days until formation of a polarized confluent layer prior to in vitro infection. Differentiation was confirmed by transepithelial resistance measurements (Millipore, Merck, Germany) surpassing 200 Ω/cm^2^. 24 h prior to experiments, culture medium was changed to antibiotic-free DMEM. A gentamicin protection assay was applied as previously described [19]. Bacteria were cultured in brain heart infusion (Merck Millipore, Germany) for four hours reaching log phase, and subsequently resuspended in prewarmed cell culture medium. Bacterial concentrations were adjusted by turbidity measurements to be added to the upper compartment at a multiplicity of infection (MOI) of 50. After 90 min of infection, medium in the upper well was removed and cells were rinsed with PBS (anprotec). Any potentially remaining extracellular bacteria were removed during 90 additional minutes of incubation in culture medium containing gentamicin (PAA, Fisher Scientific). Since gentamicin is not capable of penetrating cell membranes, extracellular bacteria in the upper well are killed exclusively. Bacterial translocation of the epithelial barrier towards the lower well was quantified using the media of the lower well with an automated bacterial cell counter (QuantomTx, BioCat, Germany) for total cells. For quantification of invasion, CaCo-2 cells were lysed with 0.2% Triton X-100 (Sigma-Aldrich, Merck, Germany) 210 min after initial infection and intracellular bacteria were counted. 

### 4.5. Plasmid Curing and PCR for hha

The plasmid was curated from Eco_b1 through prolonged incubation in ethidium bromide (EtBr 1%, AppliChem, Germany). Overnight cultures of Eco_b1 were suspended in PBS at McF of 1.0 diluted in PBS 1:10 000 and then 1:10 in LB broth (BioFroxx, Germany). EtBr was added in concentrations of 0.1–1 µg/mL and bacteria were incubated with shaking for 48 h. Vials displaying visual growth were diluted in PBS, plated on Columbia blood agar, incubated at 36 °C overnight and transferred onto both Columbia blood agar (BD, Germany) and ESBL agar (Biomérieux, Germany). After overnight incubation at 36 °C, bacterial colonies with phenotypic loss of resistance, as indicated by growth only on Columbia blood agar but not on ESBL agar, were subjected to PCR for *hha* to confirm removal of the *hha*-containing IncFII_1 plasmid. PCR for *hha* was carried out with Mytaq HS Polymerase Red (BioLine, meridian, USA) according to the manufacturer’s protocol. PCR was performed with the following primer sequences: *hha* fw ATGGCGAAAACAAAACAGGA, rev CCGGTGATTAATTCCGCTAA; and cycling conditions: 95 °C for 1 min, 30 cycles of: 95 °C for 15 sec, 60 °C for 15 sec, 70 °C for 10 sec. PCR products were visualized in 1.5% agarose gel. 

### 4.6. Gut Organoid Model

Intestinal organoids from primary human (LGR5+) stem cells were prepared as published [20]. Organoids were seeded onto collagen-coated transwells and cultured in 5% CO_2_ at 37 °C until confluency. Invasion assays were performed analogous to CaCo-2 cells.

### 4.7. Plasmid Transfer

The IncFII plasmid was transferred into the laboratory strain *E. coli* J53 via liquid mating to facilitate HGT. Overnight cultures of Eco_b1 and J53 (1:1 ratio) were cultured together in LB medium (without antibiotics) for two hours to allow for plasmid transfer and plated onto chromogenic ESBL (Sigma Aldrich, Germany) agar containing sodium azide (100 µg/mL), ceftriaxone (2 mg/mL), ceftazidime (3 mg/mL) and cefotaxime (3 mg/mL), selecting for the sodium azide and third-generation cephalosporin resistance in J53 harboring the *hha*-containing plasmid of Eco_b1 (J53pEco_b1). PCR for *hha* (see above) was used to identify successful transfer of the plasmid into selected colonies. Since the transfer of plasmid between two *E. coli* strains cannot be easily differentiated morphologically, screening for genetic identity with the donor and recipient strain of *hha*+ (by PCR) potential transconjugant was performed by WGS. 

### 4.8. Statistical Analysis and Data Presentation

Data were analyzed and presented using GraphPad Prism V5 (San Diego, CA, USA). Comparisons of the two data groups were analyzed by Mann-Whitney U test with *: *p* ≤ 0.05.

## Figures and Tables

**Figure 1 pathogens-10-01645-f001:**
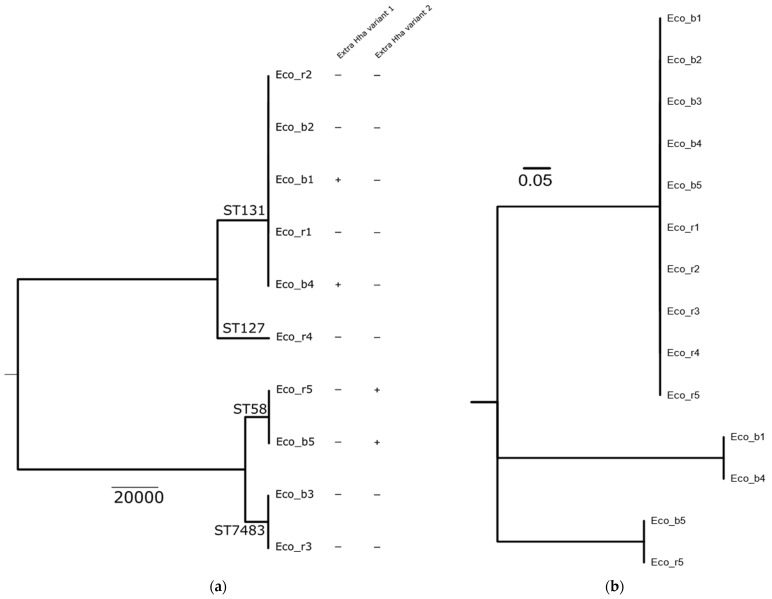
Phylogenetic tree of *E. coli* blood culture and rectal isolates of patients 1–5. (**a**) Blood culture (Eco_b) and rectal (Eco_r) isolates of P1, P2, P3 and P5 are genetically closely related. Isolates P4 are not related. In Eco_b1 and Eco_b4, the extrachromosomal *hha* variant 1 is present, while in Eco_b5 and Eco_r5, *hha* variant 2 was detected. Eco_r1, Eco_b2, Eco_r2, Eco_b3 and Eco_r3 do not possess any variant of *hha*. (**b**) Phylogenetic tree based on the protein alignment of the different Hha variant. The chromosomal Hha variant is identical in all 10 isolates.

**Figure 2 pathogens-10-01645-f002:**
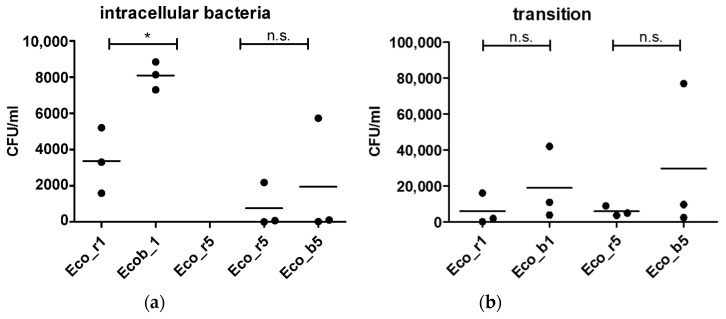
Epithelial invasion of *E. coli* isolates of P1 and P5. (**a**) Invasion assay using CaCo-2 cells in a transwell system and (**b**) permeation of bacteria into the lower chamber of the transwell. Cell layers were incubated with bacteria for 90 min and with gentamicin for another 90 min. Results are of three independent experiments. The bars depict the mean. The comparison of two data groups were analyzed by Mann– Whitney U test (one-tailed, confidence intervals 95%) with *: *p* ≤ 0.05, n.s. = not significant.

**Figure 3 pathogens-10-01645-f003:**
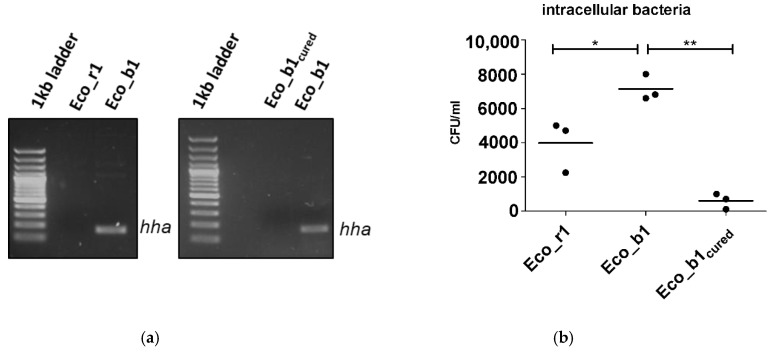
Plasmid curing of Eco_b1 and epithelial invasion of the cured strain. (**a**) *hha* PCR of Eco_r1, Eco_b1 before and after plasmid curing of Eco_b1 (Eco_b1_cured_). Invasion assay with Eco_b1 and Eco_b1_cured_ in (**b**) CaCo-2 cells, (**c**) transition into the lower chamber and (**d**) a gut organoid model. Translocation occurred into the lower well after 90 min of incubation. Results are of three or more independent experiments. The comparison of two data groups were analyzed by Mann– Whitney U test (one-tailed, confidence intervals 95%) with *: *p* ≤ 0.05, **: *p* ≤ 0.01.

**Figure 4 pathogens-10-01645-f004:**
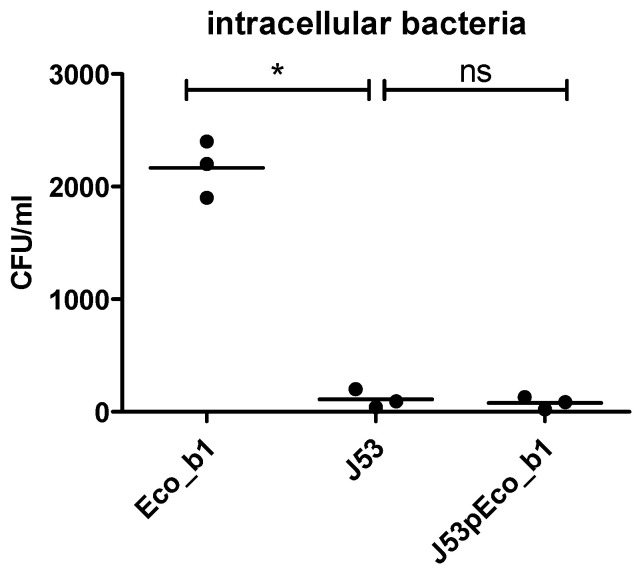
Epithelial invasion following plasmid transfer into J53. Invasion assay using the CaCo-2 model of J53 and J53pEco_b1. Results are of three independent experiments. The comparison of two data groups were analyzed by Mann– Whitney U test (one-tailed, confidence intervals 95%) with *: *p* ≤ 0.05, n.s. = not significant.

**Table 1 pathogens-10-01645-t001:** Sequencing statistics.

Isolate	Accession	MLST	Coverage	#Contigs	Largest Contig	Total Length	GC(%)	N50	N75	L50	L75
Eco_b1	SAMN20169821	131	70	81	596,711	5,127,344	50.76	218,234	124,147	8	15
Eco_b1_cured	SAMN20169822	131	39	77	596,711	5,133,246	50.76	222,766	124,147	8	15
Eco_b2	SAMN20169823	131	54	59	623,297	5,055,866	50.75	240,470	124,868	7	14
Eco_b3	SAMN20169824	7483	33	68	413,140	4,918,104	50.81	192,174	102,049	10	18
Eco_b4	SAMN20169825	131	62	73	623,463	5,139,954	50.74	222,558	135,897	7	14
Eco_b5	SAMN20169826	58	28	91	345,561	4,863,584	50.71	132,372	61,122	13	25
Eco_r1	SAMN20169827	131	59	72	596,712	5,195,130	50.71	222,897	124,147	7	15
Eco_r2	SAMN20169828	131	60	61	623,297	5,098,506	50.75	240,470	124,868	7	14
Eco_r3	SAMN20169829	7483	38	63	381,423	4,918,014	50.81	213,742	119,893	9	17
Eco_r4	SAMN20169830	127	57	42	1,600,676	5,006,799	50.42	410,665	217,547	3	7
Eco_r5	SAMN20169831	58	31	94	326,419	4,904,693	50.68	132,372	61,122	13	26
J53_Eco_b1	SAMN20169832	10	91	90	414,105	4,766,686	50.77	125,945	64,036	13	26

## Data Availability

The draft genome sequences are deposited in the NCBI GenBank database under the project number PRJNA745586.

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
