# Peer review of "Invasiveness of Escherichia coli Is Associated with an IncFII Plasmid"

_pathogens, 2021, doi:10.3390/pathogens10121645_

Round 1

Reviewer 1 Report

The authors utilize a clever approach with WGS to look at clinical isolates of E. coli and compare them to fecal samples from those patients, leveraging a paired analysis approach to attempt to determine if the presence of an hha-bearing IncFII plasmid have any relationship to invasiveness. Fix this sentence on line 59/60: In conclusion, we exemplary demonstrate the transfer of a plasmid as one mechanism 59 that may contribute to the transition of commensal to invasive E. coli strains in vivo. Exemplary is not appropriate in this sentence. While the evidence for increased invasiveness is a little weak statistically, the evidence for the reduction of invasiveness when the plasmid is cured is impressive and opens questions about the ecology of those organisms once they enter the blood stream.

n = 5 patients and 10 isolates is low for significance in this kind of study, but the novelty of the idea is strong enough that it should be published anyway. They do not show a significant increase in invasiveness between cecal (Rectal) and blood isolates, which casts some doubt on the significance their findings. They do show that curing the plasmid does reduce invasiveness. I think this paper does as they say, and submits evidence for this relationship, but is more suggestive without more examples or data. Additionally, an attempt to complement the finding did not pan out. The diversity of IncFII plasmids means that a lot more of this kind of work must be done before solid conclusion can be made about this relationship. 

Author Response

Fix this sentence on line 59/60: In conclusion, we exemplary demonstrate the transfer of a plasmid as one mechanism 59 that may contribute to the transition of commensal to invasive E. coli strains in vivo. Exemplary is not appropriate in this sentence. While the evidence for increased invasiveness is a little weak statistically, the evidence for the reduction of invasiveness when the plasmid is cured is impressive and opens questions about the ecology of those organisms once they enter the blood stream.

Response: We agree and changed the sentence into “In conclusion, our findings suggest that the transfer of a resistance plasmid may contribute to the transition of commensal to invasive E. coli strains in vivo.”.

n = 5 patients and 10 isolates is low for significance in this kind of study, but the novelty of the idea is strong enough that it should be published anyway. They do not show a significant increase in invasiveness between cecal (Rectal) and blood isolates, which casts some doubt on the significance their findings. They do show that curing the plasmid does reduce invasiveness. I think this paper does as they say, and submits evidence for this relationship, but is more suggestive without more examples or data. Additionally, an attempt to complement the finding did not pan out. The diversity of IncFII plasmids means that a lot more of this kind of work must be done before solid conclusion can be made about this relationship. 

Response: We agree that the sample size is small and should be validated in a larger cohort. We have included this limitation in the discussion. As pointed out by the reviewer the diversity of plasmids in the pathophysiology of invasive E.coli infection is not fully understood and our data, although preliminary, provide new insights into bacterial pathogenesis, which warrant further investigations.

Reviewer 2 Report

The purpose of this study was to investigate factors involved in transition of colonizing E. coli isolates to invasive strains. The authors identified E. coli blood culture isolates and clonally related isolates from rectal swabs of the same individuals through whole genome sequencing (WGS). And an IncFII plasmid carrying hha being associated with the invasiveness of one E. coli isolate was identified. This is a very interesting research. This manuscript is recommended for publication after minor revisions.

Line 63: “BSI” should be given its full name because it is the first time it appears in a paper.

Lines 69-74: The sample size from only five patients is too small. Although the authors found differences in hha genes among different patients, the results are not convincing enough.

Line 95: As can be seen from Figure 2, the repeated experiment adopted by the author was only 3 times. So it makes one wonder if the differences between the groups are due to experimental error. Figures 3 and 4 also have the same problem.

Author Response

Line 63: “BSI” should be given its full name because it is the first time it appears in a paper.

Response: Done

Lines 69-74: The sample size from only five patients is too small. Although the authors found differences in hha genes among different patients, the results are not convincing enough.

Response: We agree that the sample size is small and should be validated in a larger cohort. We have included this limitation in the discussion.

Line 95: As can be seen from Figure 2, the repeated experiment adopted by the author was only 3 times. So it makes one wonder if the differences between the groups are due to experimental error. Figures 3 and 4 also have the same problem.

Response: Thanks for the comment! We are aware that deviation in measurements can occur depending on culture conditions. However, to accommodate these variations and minimize artefacts due to experimental error, the TEER measurement was done in technical dublicates and biological triplicates.